# Chiral Nano-Liquid Chromatography and Dispersive Liquid-Liquid Microextraction Applied to the Analysis of Antifungal Drugs in Milk

**DOI:** 10.3390/molecules26237094

**Published:** 2021-11-24

**Authors:** Chiara Dal Bosco, Flavia Bonoli, Alessandra Gentili, Chiara Fanali, Giovanni D’Orazio

**Affiliations:** 1Department of Chemistry, Sapienza University of Rome, 00185 Rome, Italy; chiara.dalbosco@uniroma1.it (C.D.B.); bonoli.1705434@studenti.uniroma1.it (F.B.); alessandra.gentili@uniroma1.it (A.G.); 2Unit of Food Science and Nutrition, Department of Science and Technology for Humans and the Environment, Università Campus Bio-Medico di Roma, 00128 Rome, Italy; C.Fanali@unicampus.it; 3Istituto per i Sistemi Biologici (ISB), CNR-Consiglio Nazionale delle Ricerche, Monterotondo, 00015 Rome, Italy

**Keywords:** antifungal drugs, chiral separation, nano-LC, dispersive liquid–liquid microextraction, milk

## Abstract

A novel chromatographic application in chiral separation by using the nano-LC technique is here reported. The chiral recognition of 12 antifungal drugs was obtained through a 75 µm I.D. fused-silica capillary, which was packed with a CSP-cellulose 3,5-dichlorophenylcarbamate (CDCPC), by means of a lab-made slurry packing procedure. The mobile phase composition and the experimental conditions were optimized in order to find the optimum chiral separation for some selected racemic mixtures of imidazole and triazole derivatives. Some important parameters, such as retention faction, enantioresolution, peak efficiency, and peak shape, were investigated as a function of the mobile phase (pH, water content, type and concentration of both the buffer and the organic modifier, and solvent dilution composition). Within one run lasting 25 min, at a flow rate of approximately 400 nL min^−1^, eight couples of enantiomers were baseline-resolved and four of them were separated in less than 25 min. The method was then applied to milk samples, which were pretreated using a classical dispersive liquid–liquid microextraction technique preceded by protein precipitation. Finally, the DLLME-nano-LC–UV method was validated in a matrix following the main FDA guidelines for bioanalytical methods.

## 1. Introduction

One of the most important categories of pharmaceuticals is that of chiral drugs. Due to the difficulty and cost of synthesis, most of them are still marketed as a racemic mixture, although the pharmacological activity is often associated with only one enantiomer, with the other one being inactive or even harmful. An example of a group of chiral drugs widely used in clinical and veterinary medicine, as well as in agricultural practices, is represented by antifungal agents. They are grouped into four main chemical classes: polyene antibiotics, allylamines, fluoropyrimidines, and azole derivatives [1]. All azoles have the same mechanism of action, which consists of blocking the biosynthesis of ergosterol, a key component of the cytoplasmic membrane of fungi [2].

In farming practices, antifungal drugs are used to treat dermatophytosis or ringworm, mycotic abortions, and mycotic mastitis, especially in ruminants. Bovine mammary infections, usually caused by yeasts belonging to Candida, Cryptococcus, or Trichosporon genera, can become a chronic condition, with a negative impact on both the milk yield and quality. The great density of animals per unit area in modern housing can promote the spread of infection [3] and, therefore, high economic losses for the dairy industry. For this reason, the use of effective antifungal drugs is required. On the market, such drugs are available as racemic mixtures of various enantiomers, whose varied pharmacological activity can lower the drug’s efficacy and increase the probability of side effects [3]. Moreover, improper administration or insufficient time for drug clearance can potentially lead to the occurrence of their residues in milk, with a risk of adverse effects on consumers. Thence, the determination of these agents, potentially occurring at trace levels in livestock products, is a crucial need.

Over the last decade, several chromatographic methods have been used for the chiral and achiral separation of azole antifungal drugs, including gas chromatography (GC) [4,5], supercritical fluid chromatography (SFC) [6], liquid chromatography coupled with ultraviolet detection (LC–UV) [7,8], and liquid chromatography–mass spectrometry (LC–MS) [9,10,11,12,13]. Besides HPLC, the chiral separation of antifungal drugs has also been realized via miniaturized techniques such as nano-LC and capillary-LC (CLC) by using monolithic columns with an internal diameter (ID) of 100–150 µm; the monoliths were manufactured by in situ copolymerization using different chiral selectors: β-cyclodextrin (β-CD) [14,15], macrocyclic antibiotics, namely colistin sulfate [16], amylose 2,3 (3,5-dimethylphenylcarbamate)-6-ethylphenylcarbamate [17], and single-walled carbon nanotubes (SWCNTs) [18]. In chiral analysis, electromigration techniques are considered powerful alternatives to HPLC and GC. Thus far, the enantiomeric separation of imidazole and triazole derivatives has been achieved in capillary electrophoresis (CE), mainly by adding CDs to the background electrolyte (BGE) [19,20,21].

However, the most important issue concerning the analysis of antifungal drug residues in milk is related to the complexity of this matrix, which is, at the same time, a biological fluid and food with the characteristics of three chemical phases: emulsion, colloidal suspension, and solution [22,23]. It contains all nutrients, including proteins, lipids, and lactose, as the main components that may interfere with analytical determinations. Therefore, the sample preparation step is crucial in order to remove most interfering compounds and to concentrate the drug residues. Thus far, only two papers dealing with the determination of azole antifungals in milk have been reported. Ebrahimpour et al. realized the simultaneous extraction of miconazole and clotrimazole by using three-phase hollow-fiber liquid-phase microextraction (HF-LPME), followed by GC–flame ionization detection (FID) analysis [5]. Yahaya et al. isolated three azole antifungals from milk by ultrasound-assisted emulsification microextraction combined with dispersive micro-solid-phase extraction (USAED-µ-SPE), followed by the use of an LC diode array detector (DAD) [9]. In both cases, antifungal drugs were analyzed by achiral methods and determined as racemic mixtures.

Therefore, to the best of our knowledge, this paper is the first report on the enantiomeric determination of antifungal drugs in milk samples. To this end, lab-made silica capillary columns were packed with a polysaccharide-based stationary phase (cellulose 3,5-dichlorophenylcarbamate (CDCPC), covalently bonded to 5 µm silica particles) for the chiral separation of eight imidazole derivates (miconazole, bifonazole, butoconazole, econazole, isoconazole, ketaconazole, sertaconazole, fenticonazole) and four triazole derivates (fluconazole, posaconazole, terconazole, voriconazole). Their nano-LC–UV analysis was preceded by dispersive liquid–liquid microextraction (DLLME), a procedure that is usually applied to treat water samples and is here combined with a previous clean-up to process milk samples. The whole method, based on miniaturized extractive and separative techniques, responds to the criteria of sustainability as regards Green Analytical Chemistry.

## 2. Results and Discussion

### 2.1. Optimization of the Chromatographic Conditions

A wide variety of chiral selectors have been used to separate fungicides [24]; however, chiral stationary phases (CSPs) based on polysaccharides (i.e., cellulose and amylose) have shown excellent recognition capability, especially when derivatized with phenyl moieties, such as phenylcarbamate groups modified with electron-donating or electron-withdrawing substituents. In this study, cellulose 3,5-dichlorophenylcarbamate was tested for the chiral separation of twelve azole antifungals. The separation was carried out on a capillary column packed with a no-commercial CDCPC by means of a lab-made nano-LC system. The CSP has the helical carbon backbone with phenylcarbamate moieties as side chains, providing a highly ordered chiral conformation that is able to form a reversible complex with the molecules of the target analytes [25,26]. The chiral discrimination allowed by the CSP depends on its hydrophobic, hydrogen bonding, π-π, and dipole-type interactions [27]. The analytes selected in this study exhibit different values of hydrophobicity (see logP and hydrophilic–lipophilic balance (HLB) in Table 1) and acidity (pKa); in particular, imidazoles are weak dibasic agents whose nature has to be considered in order to achieve their chiral separation. In this framework, parameters such as the composition of the mobile phase (pH, buffer concentration, water content/organic modifier), the sample dilution solvent, and the injection volume were systematically studied by analyzing their effect on the retention factor (K), enantioresolution (Rs), chromatographic efficiency (N), and chromatographic profile. The optimal experimental conditions were obtained through a suitable compromise between the best enantioseparation and the shortest analysis time.

#### 2.1.1. Effect of pH and Buffer Solution on Enantioseparation

The analytes selected in this study had both neutral (the eight imidazole derivates) and markedly basic (the four triazole derivates) pKa values, as detailed in Table 1. Based on the chemical nature of the selector and that of the analytes, the pH of the mobile phase was expected to have an effect on their charge and, consequently, on their interactions. To investigate such aspects, aqueous solutions buffered at different pH levels were prepared—ammonium formate (pH: 3), ammonium acetate (pH 6–7), and ammonium bicarbonate (pH range 8–10)—all of them at a concentration of 5 mM. The chromatographic runs were performed in isocratic elution, using a mobile phase containing acetonitrile/buffered water (75:25, *v*/*v*). Figure 1 shows the effect on the enantiomeric resolution: as the pH increased, an improvement in chiral resolution could be observed up to pH 9, while a slight decrease appeared at pH 10. Compounds such as fluconazole, voriconazole, bifonazole, and posaconazole did not exhibit any enantiomeric separation throughout the explored pH range. With the exception of bifonazole, the basic nature of the other analytes was responsible for a strong electrostatic interaction with the CSP, which was not suitable for enantiomeric discrimination. At pH 9, the uncharged form of the neutral fungicides enhanced their weak interactions with the selector, favoring their separation. On the other hand, the same result was not observed for bifonazole, probably because its structure lacks halogen atoms or other strongly electron-attracting groups.

Considering the enantioselectivity (Rs), peak shape, and peak efficiency (data not shown), the best results were obtained at pH 9, in line with the findings of Zhang et al. [28], who obtained the baseline separation of some fungicides on a cellulose-based Chiralpak IC CSP using a HPLC system. Based on the obtained results, eight out of twelve analytes exhibited enantioselective interaction with the stationary phase under examination, namely miconazole, butoconazole, econazole, isoconazole, ketaconazole, sertaconazole, fenticonazole, and terconazole; however, only five of them were chosen as representative for further optimization experiments.

The following tests were aimed at verifying the effect of the buffer concentration in the range 2.5–40 mM. The increase in buffer concentration, and then in the mobile phase ionic strength, resulted in a general decrease in retention factors, suggesting the involvement of an ion exchange mechanism. Likewise, as shown in Figure 1b, the enantiomeric resolution was also improved, although no influence was observed for ketoconazole. It is clear that an unbuffered or low-concentration buffer mobile phase was not suitable for chiral discrimination, suggesting an important ionic interaction in diastereomeric complex formation between the CSP and the analytes. Finally, considering the better peak shape and the general improvement in peak efficiency (data not shown) for all the analytes, a buffered mobile phase of 20 mM at pH 9 was selected as the best compromise between the considered parameters to conduct further studies.

#### 2.1.2. Effect of the Mobile Phase Composition: Water Content and Organic Modifier

Keeping unchanged the buffer concentration (20 mM ammonium bicarbonate, pH 9), different ACN/H_2_O ratios were tested, varying the water content from 15% to 40% (*v*/*v*). Since no appreciable effect was observed on chromatographic efficiency, the ACN/H_2_O 85/15 (*v*/*v*) ratio was chosen as the best compromise between Rs and the lowest analysis time.

The composition of the organic modifier was changed by testing mixtures composed of ACN/MeOH in the range 0/85–85/0 (*v*/*v*) and their effect on k1 and Rs was studied. As can be seen in the k1 plot in Figure 2a, a U-shaped profile was obtained, showing a minimum for intermediate MeOH/ACN ratios. When the mobile phase was rich in MeOH or ACN, the analytes had lower solubility in the mobile phase and showed greater affinity for the stationary phase. This trend appeared to be more marked for MeOH than ACN, probably due to differences in their polarity and viscosity. The lower viscosity of ACN caused the analysis to be performed more quickly; moreover, being an aprotic solvent, its inability to form hydrogen bonds with the analytes favored their stereospecific interactions with the stationary phase (Figure 2b). Besides MeOH, other solvents (EtOH, i-prOH, and acetone) were used in combination with 90% ACN, but the best results were achieved through the individual employment of ACN as an organic modifier.

#### 2.1.3. On-Column Pre-Concentration for Sensitivity Improvement

Even if the use of a capillary column offers interesting advantages, such as higher peak efficiency and lower chromatographic dilution, the downscaling effect from a conventional chromatographic column to a capillary one requires an injection volume in the order of some tens of nanoliters. This limit, together with the reduced path length of on-column spectrophotometric detection (e.g., 75–100 µm), makes it unsuitable for trace analysis and, therefore, other analytical options such as on-column pre-concentration procedures have been studied. This approach allows one to inject a large sample volume (overloading conditions) without affecting either the column performance or the chromatographic profile. This procedure can be pursued by selecting a sample dilution solvent with an elution power that is lower than that of the mobile phase. To this end, the racemic mixtures of the five selected azole compounds (10 µg mL^−1^) were dissolved in different solvents, such as the mobile phase, acetonitrile, methanol, water, 5 mM of buffer, and 50/50 ACN/water; then, 200 nL was injected into the nano-LC system. Figure 3 shows that the best compromise in terms of low band broadening, sensitivity, and solubility was obtained with ACN/water 50/50 (*v*/*v*). In fact, high water percentages improve the chromatographic efficiency and Rs but, at the same time, cause low peak areas, probably due to reduced analyte solubility.

Afterwards, the effect of different injection volumes (range 60–800 nL) on the band wideness was investigated. To this end, the ratio between peak height (H) and peak width at half height (w_1/2_), i.e., H/w_1/2_, was plotted versus the injection volume. Generally, as the injection volume increased, a corresponding increase in the chromatographic area was observed; in addition, if the band wideness did not change, the height of the peak increased too and a linear behavior was observed. Deviations from linearity occur when the on-column focusing effect is lost; thus, the maximum injectable volume can be derived from the highest point of the linear plot. Miconazole and butoconazole did not show deviations from linearity in the studied range (data not shown), while ketoconazole, terconazole, and fenticonazole exhibited important deviations above 400 nL. Based on these results, the maximum injection volume was estimated as 300 nL.

In conclusion, among the eight compounds individually separated at the baseline (Figure 4), four of them, namely econazole, miconazole, terconazole, and ketoconazole, could be simultaneously separated by the nano-LC–UV system. For this reason and due to their widespread use in veterinary medicine, they were selected to develop a fast, simple, and green method for their determination in milk samples.

### 2.2. Optimization of the Extraction Procedure

In the optimal conditions of separation and injection, the nano-LC system was combined with an extraction procedure suitable for isolating econazole, miconazole, terconazole, and ketoconazole from milk samples. Initially, two miniaturized extractive techniques were compared: rotating-disc μ-solid-phase extraction (disc µ-SPE) based on buckypaper (BP) as the sorbent material and DLLME. To this end, preliminary experiments were carried out by using Milli-Q water (10 mL) spiked pre-extraction at 250 ng/mL with the four analytes. For each condition, three replicates were performed and recoveries were evaluated by comparison of the mean area with that obtained on a 10 mL Milli-Q water sample spiked post-extraction at the same level, according to the following equation:$$R\%=\frac{{\bar{Area}}_{pre\text{-}extraction}}{Area_{post\text{-}extraction}}\times 100$$

The extraction of the target analytes was firstly performed by means of disc µ-SPE [29,30,31,32]. Briefly, each SPE device was realized using a 25 mm diameter disc taken from a commercial BP sheet (Nanolab Inc. Waltham, MA, USA). BP is a nanostructured porous material, composed of multi-walled carbon nanotubes (MWCNTs) [33,34]. The BP disc was housed in a circle-shaped polypropylene mesh envelopment sealed at the circumference and with an open pouch to insert the disc. To ensure the disc’s rotation, a 7 mm long magnetic bar was kept in a narrow polypropylene tube fixed on the top side of the BP holder. In order to enhance the hydrophilic properties of the BP, the SPE device was immersed in a nitric acid aqueous solution (65%, *v*/*v*) for 8 h to functionalize the BP’s surface with carboxylic groups. Then, the material was rinsed with ultrapure water and MeOH, in order to remove any traces of acid, as well as any original impurities [29]. At the end of the oxidation reaction and immediately before its use, the BP device was water-conditioned (with Milli-Q) for 10 min in a 50 mL Pyrex glass beaker.

Three cleaned SPE devices were placed into 25 mL beakers, each of them containing 10 mL of water spiked with the analytes, and left under continuous magnetic stirring at 100 rpm overnight (16 h overall). Thereafter, the devices were removed using tweezers and dried thoroughly on sorbent paper. Finally, they were left under magnetic stirring (100 rpm) in 25 mL beakers containing 4 mL of MeOH, The complete desorption of the analytes took place in 20 min. The extracts were evaporated to dryness under a gentle nitrogen stream and each residue was reconstituted in 250 μL of ACN/H_2_O (85/15, *v*/*v*), centrifuged at 13,000 rpm for 10 min, and then injected for nano-LC analysis.

The results showed that only econazole was recovered satisfactorily, with a mean yield of 85%, while miconazole was isolated with a mean yield of 59%. Terconazole and ketoconazole were not recovered at all. In a previous study [33], it was verified that compounds with logP greater than 1.5 have a good affinity for BP depending on their pKa values. All analytes selected in this study had logP values in the range between 4.19 and 5.96, which does not justify the dramatic difference in the recovery values. A possible explanation was found when the hydrophilic–lipophilic balance (HLB) was taken into consideration. According to this index, compounds with an HLB value of approximately 6, such as econazole and miconazole, can be considered hydrophobic, while an HLB greater than 10, such as in the case of terconazole and ketoconazole, indicates a hydrophilic character. The elution with methanol containing 10 mM trimethylammonium chloride allowed us to recover terconazole at a percentage of less than 5%.

Owing to these unsatisfying outcomes, the recovery study was performed through a DLLME procedure by using CHCl_3_ as an extracting solvent (100, 200, and 300 µL) and CH_3_CN (500, 800 and 1000 µL) as a dispersing solvent. Among the different combinations, the best conditions were obtained with 300 µL of CHCl_3_ and 500 µL of CH_3_CN. In this case, all analytes were recovered with the following mean yields: econazole at 85%, miconazole at 100%, terconazole at 68%, and ketoconazole at 55%.

Figure 5 compares the chromatographic profile of the standard working solution with those obtained after DLLME and disc µ-SPE.

On the basis of these results, DLLME was selected for the extraction of antifungal drugs from milk samples. Based on our experience regarding the sample treatment of milk and milk derivatives [23,35,36,37,38], the protein precipitation was induced by acidified ACN. In order to obtain the maximum protein removal, screening experiments were carried out on 3 mL of milk by adding ACN in different proportions (1:2, 1:3, and 1:4 *v*/*v* milk/solvent) and by testing different types and volumes of acid in the organic solvent (acetic acid or formic acid, 100–400 μL). The best conditions were 6 mL of ACN (1:2 *v*/*v* sample/ACN) and 200 μL of acetic acid. After centrifugation, the supernatant was taken and quickly concentrated at 40 °C under reduced pressure by means of a rotavapor. The aqueous residue (approximately 3 mL) was diluted with Milli-Q water to a final volume of 10 mL, quickly filtered through 0.45 µm PTFE filters, and finally submitted to the DLLME procedure, previously optimized. Within the experimental error, recovery values from milk reflected the same percentage yields obtained from Milli-Q water.

### 2.3. Validation of the DLLME–Nano-LC–UV Method

Once optimized, the whole DLLME–nano-LC–UV method was validated considering the peak area and retention time repeatability, linear calibration range and sensitivity, LOD and LOQ, and precision and trueness.

For the intraday and interday repeatability, two blank milk samples were spiked pre-extraction at two concentration levels (level I 1.0–7.0 µg mL^−1^ and level II, 1.5–10.0 µg mL^−1^); each sample was injected six times (*n* = 6) on three consecutive days (*n* = 15). The resulting data are reported in detail in Table 2: the intraday and interday relative standard deviation (RSD%) values for retention times were in the ranges of 1.3–4.9% and 5.2–6.4%, respectively. In addition, RSD% values of peak areas ranged between 4.7% and 6.8% (intraday precision) and between 7.9% and 9.2% (interday precision), showing the repeatability of the developed chromatographic method, which is compatible with a non-thermostatic system. In addition, three columns were prepared following the same experimental conditions and tested in order to verify the reproducibility. The obtained results were quite satisfactory, with an RSD of the retention time < 7%, while retention factors were in the range of 9–12%.

Matrix-matched calibration curves were built by extracting six blank milk aliquots spiked pre-extraction with the target analytes at increasing concentrations (0.5, 1.5, 3.0, 5.0, 7.0, 10.0 µg mL^−1^).

Table 3 lists the linear dynamic range, the slope of the calibration curve (sensitivity), and the determination coefficients (R^2^), which were all higher than 0.990. LODs and LOQs were estimated by injecting extracts of blank milk aliquots spiked pre-extraction with decreasing concentrations of the target analytes until signal-to-noise ratios (S/N) of 3 and 10 were obtained, respectively. As shown in Table 3, the LODs and LOQs were in the low µg mL^−1^ range.

Precision and accuracy were estimated at two different concentration levels (in the range of 3–15 µg mL^−1^) by spiking preextraction six blank milk samples for each spike level. The experimental concentration was calculated from the calibration curves (Table 3) and then compared with the theorical one. The method’s goodness was evaluated by Student’s t test, matching the experimental t value with the tabulated one for *n* = 6 (t_5_ = 2.57 for *n* = 6, *p* = 0.05). As shown in Table 4, all experimental t values were lower than t_5_, with accuracy in the range of 89–119% and with repeatability lower than 19% (see RSD values). Although these data can be accepted (accuracy, 80–120% and RSD < 20%), importantly, the deviation could be affected by interferences in the studied matrix.

Finally, Figure 6 shows the chromatographic profiles obtained after the DLLME–nano-LC–UV analysis of a milk sample spiked with the selected compounds (a) and of a blank milk sample (b).

### 2.4. Comparison with Other Methods

To the best of our knowledge, the literature on the determination of antifungal drugs in milk and dairy samples is quite limited. Thus far, only two papers dealing with this topic have achieved higher sensitivity. For instance, by using GC–FID combined with three-phase HF-LPME, miconazole showed an LOD of 4–6 ng mL^−1^ and a recovery of 103.2–109.6. Combining USAE-D-µ-SPE based on a mesoporous carbon sorbent with HPLC-DAD analysis resulted in ketoconazole and miconazole recoveries of 83.3–111.1% and LODs in the range of 0.15–3.0 ng mL^−1^ [8]. However, although the HF-LPME method produced high sensitivity and extraction efficiency, it required a longer extraction time of up to 40 min.

The superior sensitivity achieved on a conventional LC column or GC–FID can also be explained considering the achiral chromatographic analysis of the two above-mentioned methods: for each analyte, both enantiomers co-elute, providing greater area and S/N ratios. On the other hand, the method presented here is the first application for the chiral nano-LC analysis of antifungal drugs in milk samples. This preliminary approach could be improved in the future by the development of a more sensitive hyphenation with a mass spectrometry detector.

## 3. Materials and Methods

### 3.1. Chemicals and Reagents

All chemicals were of analytical reagent RS grade and used without further purification. Acetone, acetonitrile (ACN), chloroform, ethanol (EtOH), methanol (MeOH), 2-propanol (2-PrOH), ammonium hydrogen carbonate (NH_4_HCO_3_ ≥ 99.0%, *w*/*w*), ammonia solution (30%, *w*/*w*), glacial acetic acid (99.0%, *w*/*w*), formic acid (99.0%, *w*/*w*) (FA), and trimethylammonium chloride were purchased from Aldrich−Fluka−Sigma S.r.l. (Milan, Italy). Ultrapure water was produced by a Milli-Q Plus system from Millipore (Bedford, MA, USA).

The following standard racemic mixtures (purity ≥ 97%) were obtained from Aldrich–Fluka–Sigma S.r.l. (Milano, Italy): bifonazole (1-(p,α-diphenylbenzyl)imidazole) (CAS, 60628-96-8), butoconazole ((±)-1-[4-(4-chlorophenyl)-2-[(2,6-dichlorophenyl)thio]butyl]-1H-imidazole mononitrate) (CAS, 67085-13-6), econazole (1-(2-((4-chlorophenyl)methoxy)-2-(2,4-dichlorophenyl)ethyl)-1H-imidazole) (CAS, 27220-47-9), fenticonazole ((RS)-1-[2-(2,4-dichlorophenyl)-2-hydroxyethyl]-3-[4-(phenylsulphanyl)benzyl]imidazolium nitrate) (CAS,72479-26-6), fluconazole (2-(2,4-difluorophenyl)-1,3-bis(1H-1,2,4-triazol-1-yl)propan-2-ol) (CAS, 86386-73-4), ketoconazole ((±)-cis-1-acetyl-4-(4-[(2-[2,4-dichlorophenyl]-2-[1H-imidazol-1-ylmethyl]-1,3-dioxolan-4-yl)-methoxy]phenyl)piperazine) (CAS,265-667-4), isoconazole (1-[2-(2,4-dichlorophenyl)-2-[(2,6-dichlorophenyl)methoxy]ethyl]-1H-imidazole) (CAS, 27523-40-6), miconazole ((±)-1-[2-(2,4-dichlorobenzyloxy)-2-(2,4-dichlorophenyl)ethyl]-1H-imidazole) (CAS, 22916-47-8), posaconazole (4-[4-[4-[4-[[(3R,5R)-5-(2,4-difluorophenyl)-5-(1,2,4-triazol-1-ylmethyl)oxolan-3-yl]methoxy]phenyl]piperazin-1-yl]phenyl]-2-[(2S,3S)-2-hydroxypentan-3-yl]-1,2,4-triazol-3-one) (CAS, 171228-49-2), sertaconazole (1-(2-((7-chlorobenzo[b]thiophen-3-yl)methoxy)-2-(2,4-dichlorophenyl)ethyl)-1H-imidazole) (CAS, 99592-32-2), terconazole (1-(4-((2-(2,4-dichlorophenyl)-2-(1H-1,2,4-triazol-1-ylmethyl)-1,3-dioxolan-4-yl)methoxy)phenyl)-4-(1-methylethyl)piperazine) (CAS, 67915-31-5) and voriconazole (2R,3S-2-(2,4-difluorophenyl)-3-(5-fluoropyrimidin-4-yl)-1-(1H-1,2,4-triazol-1-yl)butan-2-ol) (CAS, 137234-62-9).

Commercial buckypaper was purchased from Nanolab, Inc. (Nanolab, Waltham, MA, USA).

The standard stock solutions were prepared at a concentration of 1 mg mL^−1^, by dissolving in MeOH weighted amounts of the analytes (Ohaus DV215CD Discovery semi-micro and analytical balance, 81/210 g capacity, 0.01/0.1 mg readability) in volumetric flasks. The multi-standard working solutions (100 or 10 µg mL^−1^) were prepared by diluting the individual stock solutions with ACN/water 50/50 (*v*/*v*).

The buffer solutions (50 mL, 500 mM ammonium acetate, formate, or hydrogen carbonate) were prepared by dissolving the proper amount of ammonium salt in ultrapure water and by adjusting the pH with 5 M ammonia or formic acid. The mobile phases were daily prepared by diluting the appropriate amount of buffer solution in ACN/water mixture.

The standard stock solutions were stored in 1.5 mL Eppendorf tubes (Eppendorf S.r.l., Milano, Italy) at −18 °C. The working standard mixtures, buffer solutions, and sample extracts were stored in the dark at 4 °C prior to their use or analysis.

### 3.2. The Nano-LC System

Nano-LC experiments were performed using a semi-laboratory-assembled instrumentation composed of an Ultimate^TM^ Capillary HPLC unit from LC Packings Dionex (Amsterdam, The Netherlands), equipped with a flow splitting unit and a nano injector valve (Enantiosep, Münster, Germany; https://patents.google.com/patent/DE10260700A1/de) joined to the pump through a PEEK capillary tube (40 cm × 130 µm I.D., Vici Valco Houston, TX, USA). The injected volume was established by using the pressure-pulse-driven technique and estimated from the injection time and the column flow rate of the mobile phase [39]. Its external configuration included a 50 μL loop acting as both sample loading and mobile phase reservoir during the chiral separation. The micro-pump was used in isocratic mode delivering MeOH, while the internal split solvent was recycled as a pump solvent reservoir. The column flow rate was checked by connecting a 10 μL syringe (Hamilton, Reno, NV, USA) to the column outlet through a Teflon^®^ tube (TF-350, LC-Packing, CA, USA) and by measuring the mobile phase volume for approximately 5 min.

A UV–VIS HPE 100 BIO-RAD (Hercules, CA, USA) instrument was employed for the on-capillary UV detection. The detector was set at 195 nm; the rise time was adjusted at 0.3 s. The LC pump was manually controlled. The UV detector’s electric signal was acquired and processed by the software Chromatography Data System N2000 (Dual Channels) (BaseLine Chromtech, Zhejiang University, Republic of China).

A 75 µm I.D. capillary column, packed for 25 cm length with CDCPC 5 μm particle size, was used for the chiral separation of all the studied antifungal drugs. The chromatograms were achieved in isocratic mode by using as the mobile phase a 5 mM ammonium hydrogen carbonate ACN/H_2_O, 85/15 (*v*/*v*) mixture (pH 8.2), at a flow rate of 400 nL min^−1^.

### 3.3. Chiral Stationary Phase and Packing of Capillary Column

The chiral selector used in this study was cellulose 3,5-dichlorophenylcarbamate (CDCPC) covalently bonded to 5 µm silica particles. This material was provided by Enantiosep GmbH (Münster, Germany).

The chiral columns were prepared in our laboratory by packing the CSP into polyimide outside-coated fused-silica capillaries (375 µm O.D. and 75 µm I.D. from Polymicro TechnologiesTM, CM Scientific Ltd., Silsden, UK), according to the slurry packing method [40]. An LC series 10 HPLC pump (Perkin Elmer, Palo Alto, CA, USA) was utilized for packing the fused-silica capillary and for the fast equilibration of the capillary column. The capillary column was packed for 25 cm, while the detection window was prepared at an effective length of 26.5 cm. The total length was approximately 35 cm. The capillary was quickly conditioned with the mobile phase at 20 MPa and, thereafter, it was ready to be used in the nano-LC system. The success of the procedure, both in terms of homogeneous distribution of the stationary phase and sealing of the capillary through the localized fusion of silica, was checked by means of a Stereozoom 4 optical microscope (Cambridge Instruments, Vienna, Austria) with an illuminator.

### 3.4. Milk Samples

The whole cow pasteurized milk samples analyzed in this study were purchased in local supermarkets in Rome, Italy.

The method was validated on whole cow milk with a percentage composition in terms of proteins, carbohydrates, and fats of 3.4, 4.8, and 3.6 (*w*/*v*), respectively.

### 3.5. Sample Preparation

The analyte extraction was performed according to a two-step protocol: protein precipitation and DLLME.

The milk was initially allowed to equilibrate at room temperature; then, the carton was shaken and a 3 mL aliquot was introduced in a 15 mL polypropylene centrifuge tube (Biofil^®^, Alicante, Spain). Upon the addition of 6 mL of ACN and 200 µL of acetic acid, protein precipitation was triggered and completed, keeping the mixture under magnetic stirring for 15 min. The suspension was then centrifuged at 4000 rpm (3000× *g*) for 10 min (Universal 320R centrifuge Hettich^®^, Merck KGaA, Darmstadt, Germany), and the supernatant was transferred into a 25 mL evaporating flask to be evaporated at 40 °C with a rotavapor R-200 (Büchi Labortechnik, Flawil, Switzerland). The obtained aqueous residue (approx. 3 mL) was diluted with Milli-Q water to a final volume of 10 mL, filtered through 0.45 µm PTFE filters (Sartorius, Goettingen, Germany), and subjected to the microextraction procedure. To this end, a mixture of 500 µL of ACN (as the dispersion solvent) and 300 µL of CHCl_3_ (as the extraction solvent) was rapidly introduced into the aqueous sample by means of a 1 mL calibrated micropipette. The sample was vortex-shaken for 2 min (Reax 2000, Heidolph Instruments GmbH & CO, Germany) and then centrifuged at 4000 rpm for 5 min. After this, the chlorinated solvent containing the target analytes was withdrawn with a micro-syringe at the tube bottom. The extract (200 µL) was evaporated to dryness under a gentle nitrogen stream and the residue was reconstituted in 250 µL of ACN/H_2_O (50/50, *v*/*v*) and injected into the nano-LC system.

### 3.6. Method Validation

The DLLME–nano-LC–UV method was validated in a matrix following the main FDA guidelines for the bioanalytical methods [41].

To this end, the main evaluated parameters were recovery, precision, accuracy, sensitivity, linearity, and selectivity. All the statistical analyses were carried out using the numerical data analysis method included in Excel software (Microsoft Office 2013, Redmond, WA, USA)

## 4. Conclusions

A lab-made nano-LC–UV system has been applied for the first time to the chiral separation of some racemic mixtures of imidazole and triazole derivatives. A 75 µm I.D. capillary column packed with a CSP–polysaccharide base (cellulose 3,5-dichlorophenylcarbamate, CDCPC) allowed us to achieve a baseline enantioresolution of 8 out of 12 racemic mixtures, four of which were separated in less than 25 min at approximately 400 nL min^−1^.

These results attest to the great potential of enantioselective nano-LC, which is a rapid and green method for chiral separation, requiring small amounts of solvent and sample. Moreover, only a few milligrams of CSP material is sufficient for packing different columns. The nano-LC represents a cheaper alternative to the conventional HPLC and an interesting testbench of new, often expensive, stationary phases. This miniaturized chromatographic technique has been here combined with a miniaturized sample preparation technique (DLLME), making the developed method even greener. The application of this method to milk samples provides reliable information about the occurrence of individual stereoisomers, representing a useful tool to better understand stereoselective transfer and bioaccumulation processes.

## Figures and Tables

**Figure 1 molecules-26-07094-f001:**
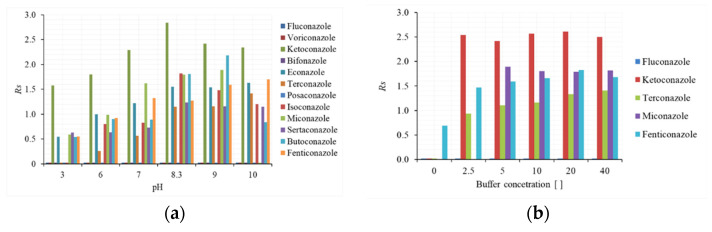
(**a**) Effect of buffer pH and (**b**) buffer concentration on enantioresolution (Rs) of some selected azole antifungal drugs. Chromatographic conditions: 75 µm I.D. packed with i-CDCPC-CPS (5 µm) Lpack = 25 cm, Leff = 26.5 cm; mobile phase, (**a**) 5 mM ammonium formate (pH 2.5 and 3.5), ammonium acetate buffer (pH 4.5 and 5.5), ammonium hydrogen carbonate (pH 8–10) in ACN/water (75:25, *v*/*v*), (**b**) ammonium hydrogen carbonate (pH 9) in ACN/water (75:25, *v*/*v*); flow rate, approximately 300–400 nL min^−1^; injected volume, approximately 100 nL; sample concentration (racemic): 100 µg mL^−1^ in ACN/water or MeOH/water (50:50, *v*/*v*); detection wavelength: 195 nm; room temperature. For other experimental conditions, see the text.

**Figure 2 molecules-26-07094-f002:**
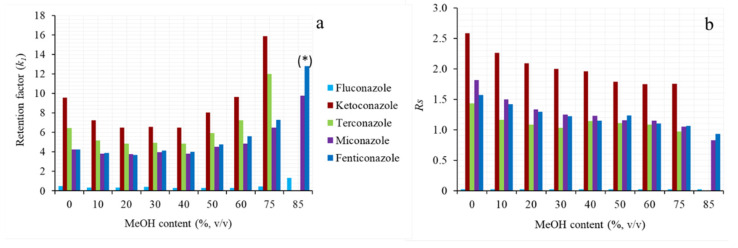
Effect of MeOH (in ACN/MeOH ratio) content on (**a**) k_1_ and (**b**) enantioresolution of some selected azole antifungal drugs. Chromatographic conditions: 20 mM ammonium hydrogen carbonate (pH 9) in organic phase/water (85/15, *v*/*v*). >90 min analysis time for ketoconazole and terconazole.

**Figure 3 molecules-26-07094-f003:**
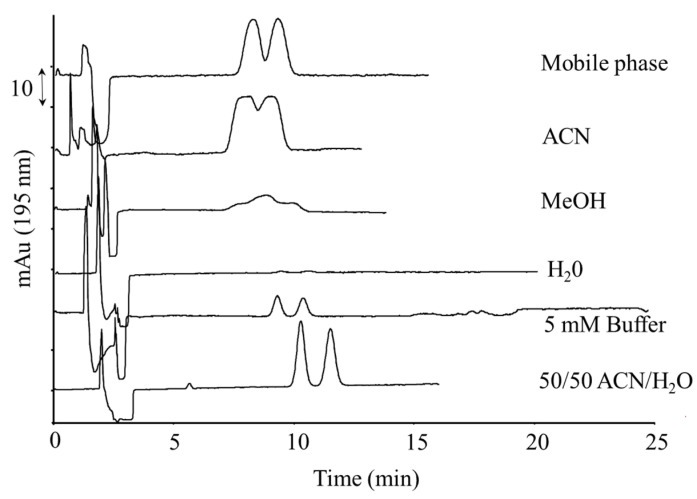
Effect of sample dilution solvent on the chromatographic profile of miconazole. Chromatographic conditions: 20 mM ammonium hydrogen carbonate (pH 9) in ACN/water (85/15, *v*/*v*); flow rate: 400 nL min^−1^; injection volume: 200 nL; sample concentration (racemic): 10 µg mL^−1^. For other experimental conditions, see Figure 1 and text.

**Figure 4 molecules-26-07094-f004:**
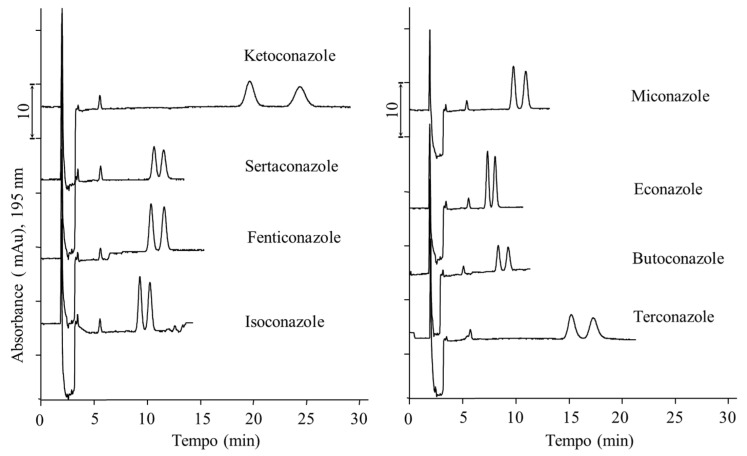
Nano-LC enantioseparation of the studied antifungal drugs. Chromatographic conditions: 20 mM ammonium hydrogen carbonate (pH 9) in ACN/water (85/15, *v*/*v*); flow rate: 400 nL min^−1^; injection volume: 300 nL; sample concentration (racemic): 10 µg mL^−1^. For other experimental conditions, see Figure 1 and text.

**Figure 5 molecules-26-07094-f005:**
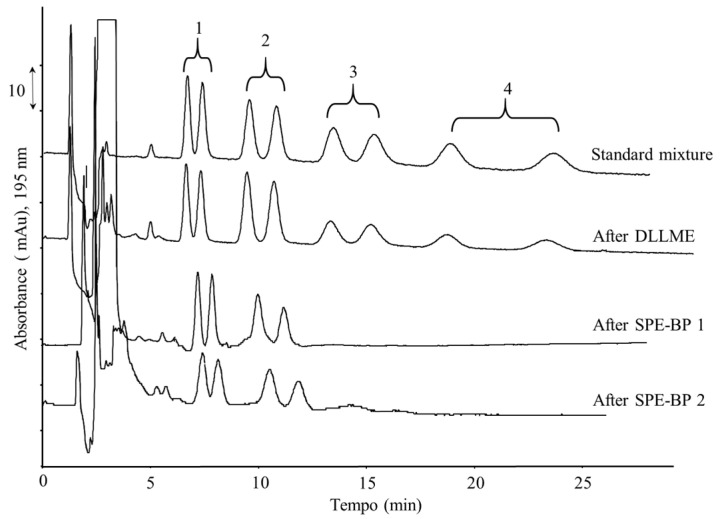
Chromatograms of Milli-Q water spiked with a multistandard solution: mix standard solution; DLLME with a mixture of ACN/CH_3_Cl (500/300 μL) as the dispersing and extracting solvent, respectively; SPE-BP 1, with desorbing solvent MeOH; SPE-BP 2 with desorbing solvent [(CH_3_)_4_N]Cl hydrate 10 mM MeOH. Experimental conditions: 1—econazole, 2—miconazole, 3—terconazole, 4—ketoconazole.

**Figure 6 molecules-26-07094-f006:**
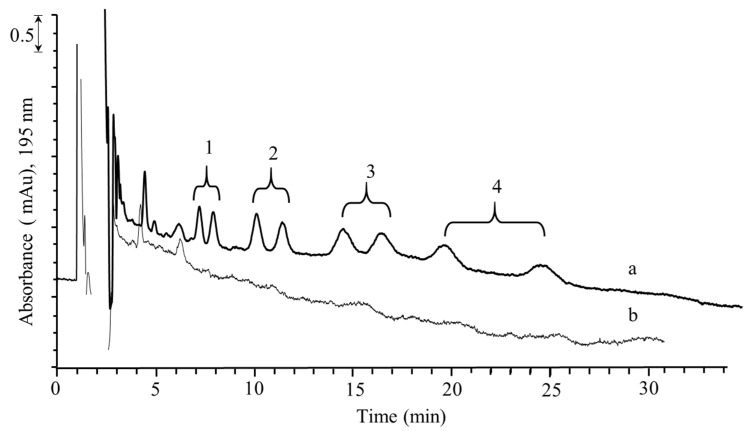
UV chromatograms of milk samples after DLLME–nano-LC–UV method: (a) spiked with selected antifungal drugs, (b) blank sample. Experimental conditions: spiked racemic mixture 5–15 µg/mL ng. For further information, see Figure 5 and text.

**Table 1 molecules-26-07094-t001:** pKa, logP, and HLB values of the studied antifungal drugs.

Analyte	pKa	LogP ^(*)^	HLB ^(*)(**)^
Fluconazole	12.68	0.56	9.56
Voriconazole	12.7	1.82	8.24
Ketoconazole	6.42	4.19	10.39
Bifonazole	6.36	5.23	2.66
Econazole	6.48	5.35	5.77
Terconazole	8.45	5.37	14.26
Posaconazole	14.85	5.41	8.15
Isoconazole	6.48	5.96	6.01
Miconazole	6.48	5.96	6.01
Sertaconazole	6.48	6.23	5.93
Butoconazole	6.51	6.55	5.52
Fenticonazole	6.48	6.94	4.16

(*) MarvinSketch 19.24.0 (hpp://www.chexon.com). (**) HLB, hydrophilic–lipophilic balance).

**Table 2 molecules-26-07094-t002:** Peak area repeatability and reproducibility of spiked milk samples obtained with DLLME–nano-LC–UV method.

		Peak Area (RSD, %)			
		Level I (3.0–7.0 µg mL^−1^)		Level II (6.0–15.0 µg mL^−1^)	
Peak		Intraday (*n* = 6)	Interday (*n* = 15)	Intraday (*n* = 6)	Interday (*n* = 15)
#1	Econazole	6.4	9.2	6.5	9.0
#2	Econazole	6.7	8.8	6.8	8.8
#1	Miconazole	5.2	7.3	6.4	7.5
#2	Miconazole	4.7	7.8	6.1	8.4
#1	Terconazole	4.9	6.9	4.7	7.3
#2	Terconazole	5.2	7.2	5.5	7.5
#1	Ketoconazole	6.0	8.3	5.9	8.4
#2	Ketoconazole	5.7	8.8	6.7	9.1

**Table 3 molecules-26-07094-t003:** Linear dynamic range for the enantiomers of the selected target compounds in spiked milk samples after the sample pretreatment DLLME and analyzed by nano-LC–UV.

		Linear Dynamic Range (µg mL^−1^)	Regression Equation (*n* = 6)				
			y = mx + n				
Peak #	Analyte		M ± t·S_m_ (10^4^)	N ± t·S_n_ (10^4^)	*R* ^2^	LOD^a^_method_ (µg mL^−1^)	LOQ^b^_method_ (µg mL^−1^)
1	Econazole	0.50–10.0	1.9 ± 0.8	1.5 ± 1.6	0.990	0.05	0.14
2		0.50–10.0	2.0 ± 1.0	1.2 ± 1.4	0.992	0.06	0.16
1	Miconazole	1.5–10.0	2.3 ± 1.1	1.3 ± 1.4	0.991	0.20	0.80
2		1.5–10.0	2.2 ± 1.2	1.1 ± 1.2	0.990	0.30	1.10
1	Terconazole	4.0–20.0	1.8 ± 0.8	1.5 ± 1.7	0.990	1.10	3.20
2		4.0–20.0	1.7 ± 1.1	1.7 ± 1.8	0.991	1.20	3.50
1	Ketokonazole	7.0–20.0	1.8 ± 1.2	1.1 ± 1.3	0.991	2.50	6.10
2		7.0–20.0	1.9 ± 1.3	0.9 ± 1.2	0.992	2.70	6.80

m, slope; Sm, standard deviation of the slope; n, intercept; Sn, standard deviation of the intercept; R^2^, determination coefficient. (a) Calculated as the concentration associated with an S/N ratio of 3; (b) Calculated as the concentration associated with an S/N ratio of 10. t = (Student’s t test) = 2.78, α = 0.05.

**Table 4 molecules-26-07094-t004:** Results of the precision and accuracy study of the DLLME–nano-LC–UV method on spiked milk samples. For experimental conditions, see Materials and Methods section.

Analytes	Peak #	Spiked Level (µg mL^−1^)	Found ^(a)^ (µg mL^−1^)	Accuracy (%), (RSD,%)	t
Econazole	1	3.0	2.7 ± 0.3	91 (19)	0.69
	2	5.0	5.2 ± 0.6	104 (12)	1.46
	1	3.0	2.8 ± 0.3	92 (17)	1.98
	2	5.0	5.4 ± 0.2	108 (15)	1.65
	1	3.0	2.7 ± 0.3	89 (12)	0.66
Miconazole	2	5.0	4.8 ± 0.4	95 (17)	0.37
	1	3.0	2.8 ± 0.8	93 (14)	0.87
	2	5.0	5.0 ± 1.4	99 (18)	0.22
	1	5.0	5.2 ± 0.8	104 (12)	1.01
Terconazole	2	10.0	9.5 ± 1.4	95 (17)	2.27
	1	5.0	5.4 ± 1.1	108 (15)	2.39
	2	10.0	9.2 ± 1.5	92 (19)	1.59
	1	7.0	8.1 ± 1.4	115 (12)	0.01
Ketokonazole	2	15.0	15.6 ± 2.8	104 (14)	2.51
	1	7.0	8.3 ± 1.6	119 (14)	0.90
	2	15.0	16.2 ± 3.1	108 (16)	1.93

(a). Average value ± confidence interval (five determinations, 95% confidence value) t = (Student’s t test) = 2.56, α = 0.05.

## Data Availability

Not applicable.

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
