# Peer review of "Chiral Nano-Liquid Chromatography and Dispersive Liquid-Liquid Microextraction Applied to the Analysis of Antifungal Drugs in Milk"

_molecules, 2021, doi:10.3390/molecules26237094_

Round 1

Reviewer 1 Report

The manuscript describes chiral separation and analysis of antifungal drugs in milk by nano liquid chromatography. The topic with miniaturized LC is big essential and The obtained results are looking promising. This reviewer only suggest to re-consider to the authors following point before publication; 

Page 17, Line 540-541, Ref. 39 seems not suitable. 

Author Response

The manuscript describes chiral separation and analysis of antifungal drugs in milk by nano liquid chromatography. The topic with miniaturized LC is big essential and The obtained results are looking promising. This reviewer only suggest to re-consider to the authors following point before publication;

Page 17, Line 540-541, Ref. 39 seems not suitable.

We agree with the reviewer and we modified the reference in:

[39] Manz, A; Simon, W. Injectors for open-tubular column liquid chromatography with 106 theoretical plates at retention times in the minute range. J Chromatogr.A 1987, 387, 187-196.

(x) English language and style are fine/minor spell check required

We revised the English language and style throughout the manuscript as suggested by the reviewer.

Reviewer 2 Report

This paper describes a method for the chiral separation of antifungal compounds in milk by nano-liquid chromatography. Dispersive liquid-liquid microextraction has been applied for sample treatment. In my opinion, the paper contains valuable results and deserves publication, although major revision is needed before being published. These are the main points to be considered:

  • The paper is unnecessarily long. For example, the introduction contains information that is not relevant considering the goal of the study (such as categories of drugs or farming practices), or is too basic (like the fact that different enantiomers may cause different effects on the target). I suggest a substantial condensation of this section. Figure 1 (last paragraph) is missing. I recommend including that figure as a supplementary content.

  • Section 2.1 should be also condensed. In my opinion, part of the results can be considered part of preliminary studies, as evidenced by the fact that only four compounds have been selected to develop the method. Again, a substantial part of this section should be shown as supplementary content; only the main conclusions (optimum conditions) must be presented. Besides, there are too many vague explanations such as “suggesting the involvement of an ion-exchange mechanism”, “suggesting an important ionic interaction…”,  “probably due to..”, probably because of…”. ). I suggest combining in a single figure current figures 1, 2 and 3 and to move current figure 4 to the supplementary content, and eliminating from the discussion non verified results.

  • Since the nano liquid chromatographic system was home-made, I think that an study on the repeatability of the retention times is needed, particularly taking into account that this parameter is critical in chiral analysis. Please, add data on the retention time variations in successive injections, in different days and in different columns.

  • Because of the unacceptable recoveries obtained, it is clear that µ-SPE is not a good option in this case. Therefore, this part of the study (paragraphs 2nd, 3rd and 4th of section 2.2) has to be removed, and the paper has to be rewritten accordingly (Figure 6, and experimental). By the way, even though only four compounds were finally included in the study with milk samples, the chromatograms of figures 6 and 7 clearly indicate that gradient elution would have been a better option for this separation. This should be discussed in the text.    

  • Section 2.3: I do not understand why the authors refer two different concentration levels (1.0-7.0 and 1.5-10.0 µg/mL). From my understanding, both concentration levels are quite similar. In view of the dynamic ranges of Table 2, I think that the repeatability on peak areas needs to be tested at concentrations <≈5 and >≈7,5 µg/mL for econazole and miconazole, and <≈5 and >≈15 µg/mL for the other two compounds (as the authors did in Table 3).

  • Table 3: accuracy data should be expressed as relative error (single value). The values obtained for miconazole and ketokonazole seem to be biased. This point needs to be addressed.

Author Response

This paper describes a method for the chiral separation of antifungal compounds in milk by nano-liquid chromatography. Dispersive liquid-liquid microextraction has been applied for sample treatment. In my opinion, the paper contains valuable results and deserves publication, although major revision is needed before being published. These are the main points to be considered:

The paper is unnecessarily long. For example, the introduction contains information that is not relevant considering the goal of the study (such as categories of drugs or farming practices), or is too basic (like the fact that different enantiomers may cause different effects on the target). I suggest a substantial condensation of this section.

We have simplified the introduction as suggested by the reviewer.

 Figure 1 (last paragraph) is missing. I recommend including that figure as a supplementary content.

Thank you for your comment. Since the molecular structure are known, we prefer to remove this figure 1.

Section 2.1 should be also condensed. In my opinion, part of the results can be considered part of preliminary studies, as evidenced by the fact that only four compounds have been selected to develop the method. Again, a substantial part of this section should be shown as supplementary content; only the main conclusions (optimum conditions) must be presented.

We have condensed section 2.1 as much as possible, according to the reviewer suggestion. However, when chiral compounds are separated by using a new generation chiral stationary phase, the study of the chromatographic conditions is an important part of the experimental work, therefore we believe that it should be described in the research paper.. Many chiral HPLC and especially nano-LC research works focused on the study of the stationary phase analyze the enantiorecognition towards one or more selected classes of compounds in different experimental conditions: reverse phase, normal phase, polar organic phase, etc. Here, the trends and influences on peak efficiency, retention factors and enantioresolution are highlighted.

Besides, there are too many vague explanations such as “suggesting the involvement of an ion-exchange mechanism”, “suggesting an important ionic interaction…”,  “probably due to..”, probably because of…”. ).

We believe that hypothetical sentences are more appropriate in cases like this, where many chiral mechanisms are still unknown and just some predictions can be made in the absence of previous supporting literature. Our group is still working on the study of this new stationary phase for achieving a deep understanding of the several interaction mechanisms involved in the chiral separation under different experimental conditions.

I suggest combining in a single figure current figures 1, 2 and 3 and to move current figure 4 to the supplementary content, and eliminating from the discussion non verified results.

As suggested by the reviewer, figures 1 and 2 have been combined in a single figure since they both represented the effect of the buffer solution (pH and concentration). For clarity of presentation Figure 3 has not been merged since it shows the effect of organic phase MeOH or ACN, where other interactions are involved. We think that figure 4 should be shown in the paper because it represents the dramatic effect of the solvent sample dilution on the chromatographic profile, which is an important aspect in miniaturized techniques, nano/capillary-LC and capillary electrochromatography, while it is completely negligible in HPLC experiments. Since the effect of the solvent dilution depends on the stationary phase and on the target compounds, its study is important for obtaining the best chromatographic profile and the highest sensitivity in overloading conditions.

Since the nano liquid chromatographic system was home-made, I think that an study on the repeatability of the retention times is needed, particularly taking into account that this parameter is critical in chiral analysis. Please, add data on the retention time variations in successive injections, in different days and in different columns.

In the section 2.1 in table 2 (ex table 1) all repeatability data were reported: intra and inter days repeatability. Some additional experiments were performed for evaluating the batch-to-batch column repeatability. We have added these new results in this section.

Because of the unacceptable recoveries obtained, it is clear that µ-SPE is not a good option in this case. Therefore, this part of the study (paragraphs 2nd, 3rd and 4th of section 2.2) has to be removed, and the paper has to be rewritten accordingly (Figure 6, and experimental).

We agree with the reviewer that the result of the micro-SPE by using CNTs is negative. However, in our opinion this result is still useful for understanding which kind of interactions take place between the analytes and the sorbent material. In the text, we raised the hypothesis that log P is not sufficient to explain the extraction capacity of CNTs. In this regard, the HLB parameter should be also considered for further evaluation in the sample preparation.

By the way, even though only four compounds were finally included in the study with milk samples, the chromatograms of figures 6 and 7 clearly indicate that gradient elution would have been a better option for this separation. This should be discussed in the text.

In chiral liquid chromatography it is not appropriate to use gradient elutions. The reversible diasteroisomeric complex between the stationary phase and the analyte happens for multiple interactions and then the chromatographic elution mechanism is multimodal. For instance, the change of the organic solvent content produces unpredictable effects on both the retention factor and the chiral resolution, contrary to what happens in non-chiral RP- HPLC.

Section 2.3: I do not understand why the authors refer two different concentration levels (1.0-7.0 and 1.5-10.0 µg/mL). From my understanding, both concentration levels are quite similar. In view of the dynamic ranges of Table 2, I think that the repeatability on peak areas needs to be tested at concentrations <≈5 and >≈7,5 µg/mL for econazole and miconazole, and <≈5 and >≈15 µg/mL for the other two compounds (as the authors did in Table 3).

We thank the reviewer for reporting this mistake. We changed the manuscript in the table 2.

Table 3: accuracy data should be expressed as relative error (single value).

The accuracy value is reported as % in column 5, while its error is expressed in RSD in brackets.

The values obtained for miconazole and ketokonazole seem to be biased. This point needs to be addressed.

We apologize to the reviewer for not fully understanding his remark.

We have modified the text to justify the limit value of the accuracy for ketoconazole (about 120%), which can be related to the interference of the matrix.

“Although these data can be accepted (accuracy, 80-120% and RSD <20%), the important deviation could be affected by interferences in studied matrix.”

Reviewer 3 Report

The manuscript reports an analytical method based on dispersive liquid-liquid microextraction and chiral nano-liquid chromatography for the analysis of antifungal drugs in milk. The authors performed chromatographic optimization, optimization of the extraction procedure and validation of the created method. From analytical point of view, the study has some novel aspects. The authors highlighted the advantages of their study in the introduction. Indeed, they used lab-made silica capillary columns, dispersive liquid-liquid microextraction (DLLME), a miniaturized extraction and separative techniques that follow the principles of Green Analytical Chemistry. The manuscript is well-written but lacks the application in real samples.

I suggest acceptance of the manuscript after the authors apply their method in few samples to demonstrate the fitness for purpose of their method.

Author Response

The manuscript reports an analytical method based on dispersive liquid-liquid microextraction and chiral nano-liquid chromatography for the analysis of antifungal drugs in milk. The authors performed chromatographic optimization, optimization of the extraction procedure and validation of the created method. From analytical point of view, the study has some novel aspects. The authors highlighted the advantages of their study in the introduction. Indeed, they used lab-made silica capillary columns, dispersive liquid-liquid microextraction (DLLME), a miniaturized extraction and separative techniques that follow the principles of Green Analytical Chemistry. The manuscript is well-written but lacks the application in real samples.

I suggest acceptance of the manuscript after the authors apply their method in few samples to demonstrate the fitness for purpose of their method.

It’s difficult to evaluate if the method fits the purpose from the results obtained on real samples, because, according to the published literature, azole antifungals have never been detected in milk real samples even when more sensitive methods were used (Othman, N.; Lim, V.; Ramachandran, M. R.; Sanagi, M. M.; Kamaruzaman, S.; Hirota, Y.; Norikazu N.; Noorfatimah Y.; Yahaya, N. Rapid ultrasound-assisted emulsification microextraction combined with COU-2 dispersive micro-solid phase extraction for the determination of azole antifungals in milk samples by HPLC-DAD. Chromatographia 2017, 80, 1553-1562). It can depend on the insufficient sensitivity of the available methods or on the high-quality standards of the commercial milk. We consider this experimental work as a preliminary analytical method for developing a more sensitive and specific one by using MS detector. In addition, as reported in the conclusions: “These results attest the great potential of enantioselective nano-LC, which is a rapid and green method for chiral separations, requiring small amounts of solvent and sample”

(x) English language and style are fine/minor spell check required

We revised the English language and style throughout the manuscript as suggested by the reviewer.

Round 2

Reviewer 2 Report

The manuscript has been substantially improved. I recommend acceptance.